# Saturated and Polyunsaturated Fatty Acids Production by *Aurantiochytrium limacinum* PKU#Mn4 on *Enteromorpha* Hydrolysate

**DOI:** 10.3390/md21040198

**Published:** 2023-03-23

**Authors:** Yaodong He, Xingyu Zhu, Yaodong Ning, Xiaohong Chen, Biswarup Sen, Guangyi Wang

**Affiliations:** 1Center of Marine Environmental Ecology, School of Environmental Science and Engineering, Tianjin University, Tianjin 300072, China; 2School of Fishery, Zhejiang Ocean University, Zhoushan 316022, China; 3Key Laboratory of Systems Bioengineering (Ministry of Education), Tianjin University, Tianjin 300072, China; 4Qingdao Institute for Ocean Technology of Tianjin University Co., Ltd., Qingdao 266237, China

**Keywords:** *Enteromorpha*, hydrolysate, thraustochytrids, fermentation, fatty acids, reducing sugars

## Abstract

Thraustochytrids are unicellular marine heterotrophic protists, which have recently shown a promising ability to produce omega-3 fatty acids from lignocellulosic hydrolysates and wastewaters. Here we studied the biorefinery potential of the dilute acid-pretreated marine macroalgae (*Enteromorpha*) in comparison with glucose via fermentation using a previously isolated thraustochytrid strain (*Aurantiochytrium limacinum* PKU#Mn4). The total reducing sugars in the *Enteromorpha* hydrolysate accounted for 43.93% of the dry cell weight (DCW). The strain was capable of producing the highest DCW (4.32 ± 0.09 g/L) and total fatty acids (TFA) content (0.65 ± 0.03 g/L) in the medium containing 100 g/L of hydrolysate. The maximum TFA yields of 0.164 ± 0.160 g/g DCW and 0.196 ± 0.010 g/g DCW were achieved at 80 g/L of hydrolysate and 40 g/L of glucose in the fermentation medium, respectively. Compositional analysis of TFA revealed the production of equivalent fractions (% TFA) of saturated and polyunsaturated fatty acids in hydrolysate or glucose medium. Furthermore, the strain yielded a much higher fraction (2.61–3.22%) of eicosapentaenoic acid (C20:5n-3) in the hydrolysate medium than that (0.25–0.49%) in the glucose medium. Overall, our findings suggest that *Enteromorpha* hydrolysate can be a potential natural substrate in the fermentative production of high-value fatty acids by thraustochytrids.

## 1. Introduction

Marine microalgae have great biotechnological applications because of their capacity to synthesize a wide variety of valuable metabolites [1,2,3]. They are considered the most promising alternative sources of feed, biofuels, and chemicals [4,5]. Some strains of Stramenopile are well-known for their ability to produce high amounts of polyunsaturated fatty acids (PUFAs) and saturated fatty acids (SFAs), which find applications in nutraceuticals and biofuels industries [6]. Particularly, thraustochytrids, which are unicellular heterotrophic marine protists of the Stramenopile group, often considered as non-photosynthetic microalgae [7], can utilize several carbon sources (e.g., glucose, glycerol, fructose, sucrose, and biomass hydrolysate) for their growth and fatty acid production [8,9]. However, the production of fatty acids by thraustochytrids is largely affected by the carbon source of the fermentation medium [9]. The type and concentration of the carbon source can greatly influence the growth and metabolism of thraustochytrids [9,10], and ultimately affect the quantity and quality of the fatty acids produced. For example, some studies have shown that thraustochytrids can produce higher levels of certain types of fatty acids when grown on specific carbon sources, such as glucose or glycerol, compared to other carbon sources [9,11]. In addition, the concentration of the carbon source can also play a role in determining the fatty acid production by thraustochytrids, with some studies indicating that higher concentrations of glycerol can result in higher lipid accumulation and higher levels of certain fatty acids [11]. Overall, the carbon source used in the fermentation medium can have a significant impact on the production of fatty acids by thraustochytrids, and the careful selection of the appropriate carbon source and concentration is crucial for optimizing the yield and quality of the fatty acids produced.

Although heterotrophic cultivation of thraustochytrids yields substantial productivity, it requires large nutrient input. Moreover, fermentation of commercial substrates can significantly increase the overall cost of the bioprocess for fatty acid production using thraustochytrids [12]. To this end, natural substrates should be carefully selected to make heterotrophic cultivation of thraustochytrids feasible [13]. So far, thraustochytrids have been reported to use certain natural substrates such as bagasse [14,15], Jerusalem artichoke [16], sweet sorghum straw juice [17], crude glycerol [18,19,20], mixed wastewater [21], cane molasses [22], hemp [23], and forest biomass [24,25] for the production of fatty acids. Compared to lignocellulosic hydrolysates, the macroalgal hydrolysate contains a relatively low amount of lignin and cellulose, and their pretreatment and saccharification are much simpler and easier [26]. However, the chemical composition of macroalgae biomass is heavily influenced by the species and the conditions under which it grows. Macroalgae usually have a dry matter content of 10 to 25% by weight [27], and the majority of this consists of carbohydrates, which can make up to 60% of the total weight [28]. While all macroalgae have cellulose, the specific types of carbohydrates present vary among the different groups. Brown algae, for instance, mainly contain laminarin, mannitol, alginate, glucan, and fucoidan, while red algae have carrageenan, agar, and lignin, and green algae possess mannan, ulvan, and starch [29]. To achieve environmental benefits and improve product quality while eliminating any residual waste, the most desirable approach for utilizing marine macroalgae waste appears to be the cascade biorefinery approach [30].

A previous research explored the use of macroalgae waste as a resource for food and chemicals [31]. A single-step microwave process was used to produce a growth medium for microbial fermentation from a variety of UK native seaweeds, with the brown seaweeds, particularly kelp, showing the highest potential. The oleaginous yeast *Metschnikowia pulcherrima* was used to metabolize the medium and produce lipids, achieving a yield of 6.9 g/L yeast biomass containing 37.2% (*w*/*w*) lipid. This system provided a low-cost route to edible microbial oils and renewable feedstock for oleochemicals. However, despite the potential benefits of using marine algal biomass as a substrate for fatty acid production, there has been limited research on this topic. For instance, the potential of marine algal biomass as a substrate for fatty acid production using thraustochytrids has seldom been studied. Further studies are needed to evaluate the feasibility and viability of using marine algal biomass for fatty acid production and to optimize the process for maximum efficiency and yield.

*Enteromorpha*, a marine green macroalgal genus, has recently drawn the attention of researchers for biorefinery applications. *Enteromorpha* has important ecological and economic implications, as it serves as a food source and habitat for many marine organisms [32,33]. However, excess growth of *Enteromorpha* can also lead to negative environmental impacts, such as oxygen depletion and habitat degradation. They occur from the intertidal to the upper subtidal zones of the world’s ocean and are the most common fouling algae [34]. Due to climate change and coastal eutrophication, *Enteromorpha* blooms occur almost every year along the coast of Qingdao, China, leading to a series of environmental and economic losses [32,35]. Conventional methods of controlling fouling *Enteromorpha* involve the use of harmful substances such as oxidants, acids, heavy metal compounds, and synthetic fungicides, which can be applied directly to water or to *Enteromorpha*-infested surfaces or incorporated into marine coatings as antifouling agents [36]. Despite being cost-effective and convenient, these methods pose significant environmental challenges. The use of hydrogen peroxide and sulfuric acid solutions to eradicate algae propagules, for instance, can result in water contamination and adversely impact other marine organisms. Therefore, continued research is needed to develop effective strategies for utilizing their biomass and indirectly controlling its negative impacts on marine ecosystems.

The biomass of *Enteromorpha* has been typically discarded as an environmental waste [32]. Interestingly, their biomass is composed of many sulfated polysaccharides containing glucose, xylose, glucuronic acid, galactose, and rhamnose [34,37,38,39]. These polysaccharides show many bioactive functions such as anticoagulant activity, antioxidant activity, antitumor activity, and immunomodulatory activity [33,34,40,41]. Furthermore, the *Enteromorpha* hydrolysate has been reported in several studies as a potential substrate for production of biofuels and bioactive compounds [42,43,44,45]. Some studies also suggested the potential application of *Enteromorpha* in bioremediation of heavy metals [46,47]. However, no effort has been made to ferment *Enteromorpha* hydrolysates to produce high-value fatty acids using thraustochytrids. 

In this study, the content and composition of total reducing sugars in the hydrolysate derived from the dilute acid hydrolysis of the dried *Enteromorpha* were analyzed for their potential as a substrate for biomass and fatty acid production using a previously isolated thraustochytrid strain (*Aurantiochytrium limacinum* PKU#Mn4). Our study provides the first report on the potential application of *Enteromorpha* hydrolysate as natural substrate to produce SFAs and PUFAs via microbial fermentation.

## 2. Results and Discussion

### 2.1. Sugar Content and Composition of Enteromorpha Hydrolysate

The total sugar content of *Enteromorpha* hydrolysate was 43.9% ± 1.9%, accounting for almost half the dried weight of *Enteromorpha.* The high sugar content of *Enteromorpha* hydrolysate could make it a suitable substrate for lipid fermentation by thraustochytrids. Further analysis of the total sugars indicated that glucose was the major sugar in the hydrolysate (Appendix A). A previous sugar compositional analysis of *Enteromorpha compressa* revealed that it consists of a high amount of rhamnose along with a smaller quantity of other monosaccharides such as glucose, xylose, and galactose [37]. Likewise, another study found glucose, xylose, rhamnose, and glucuronic acid as the main reducing sugars in *Enteromorpha* hydrolysate [38]. The extraction and hydrolysis methods in previous studies were important factors that determined the sugar composition of *Enteromorpha.* For instance, when 20 and 50 g/L of *Enteromorpha* polysaccharides were hydrolyzed with the crude enzymes of *Vibrio* sp. H11, the concentrations of reducing sugars were 0.8 g/L and 0.93 g/L, respectively [48]. These concentrations were 49.03% and 36.12% higher than that of the control. To assess the potential of *Enteromorpha* sp. as a bioenergy resource, another study examined the impact of reaction conditions such as solid-to-liquid ratio, reaction temperature, and reaction time on sugars produced through a combined process of hydrothermal and enzymatic hydrolysis [49]. The results showed that under specific conditions of solid-to-liquid ratio of 1:10, reaction temperature of 170 °C, and reaction time of 60 min, hydrothermal hydrolysis produced 7.3 g/L (8% yield) of total reducing sugar. Subsequent enzymatic hydrolysis of samples treated at 170 °C for 30 min resulted in a yield of 20.1 g/L (22%). In yet another study, a method to use the biomass of invasive brown alga *Rugulopteryx okamurae* to produce monosaccharides was reported, which can then be used to create valuable bioproducts through fermentation [50]. The method involved using *Aspergillus awamori* in a solid-state fermentation process to pretreat the seaweed before enzymatic hydrolysis. The study found that a five-day pretreatment with *A. awamori* followed by 24 h of enzymatic saccharification was the most effective condition, resulting in the production of approximately 240 g of total reducing sugars per kg of dried biomass, with glucose being the primary sugar obtained.

Interestingly, our study demonstrated that the use of diluted sulfuric acid for biomass hydrolysis can be beneficial for improved substrate utilization by thraustochytrids during fermentation.

### 2.2. Lipid Production Potential of Enteromorpha Hydrolysate 

To test the potential of the *Enteromorpha* hydrolysate as a fermentation substrate for lipid production by the PKU#Mn4 strain, the cell growth, substrate consumption, and intracellular lipid content were compared between cultivation on hydrolysate (60 g/L) and glucose (25 g/L) media. The estimated maximum growth rate of the strain was 6.20 gL^−1^d^−1^ and 4.18 gL^−1^d^−1^ on the hydrolysate and glucose media, respectively (Table 1). However, the strain reached the stationary phase relatively earlier (ca. 36 h) in the hydrolysate medium than in the glucose medium (48 h) (Figure 1a). These results indicated possible growth inhibition of the PKU#Mn4 strain on hydrolysate medium after rapid consumption of the fermentable sugars. Our findings are in agreement with previous studies that highlighted the potential growth inhibition by certain compounds, e.g., furfural [51] and polyphenols [52], which are often generated in the acid hydrolysis process of biomass [53,54]. Interestingly, despite growth inhibition, the DCW of the strain on the hydrolysate medium reached an estimated maximum value of 3.635 g/L, which was 66.47% of the DCW (5.469 g/L) on the glucose medium (Table 1). 

The consumption pattern of total reducing sugars in the *Enteromorpha* hydrolysate was similar to that of the glucose medium (Figure 1b). However, the sugar consumption by the PKU#Mn4 strain started to relatively retard after 24 h of cultivation on the hydrolysate medium. At the end of fermentation, the residual reducing sugars content in the hydrolysate medium was 8.83 ± 0.37 g/L, which indicated that about 63.5% of the total reducing sugars was fermentable. At the same time, most of the glucose (96.7%) was consumed when the strain was cultivated on a glucose medium. *Enteromorpha* hydrolysates are reported to constitute reducing sugars other than glucose [38], and some of these sugars (xylose, rhamnose, and glucuronic acid) are generally not amenable to fermentation by thraustochytrids. 

The time course of fatty acid accumulation by the PKU#Mn4 strain on the hydrolysate and glucose media exhibited a sigmoid pattern (Figure 1c). The modified Gompertz model fitted the experimental data satisfactorily (Table 1, Figure 1c). The estimated maximum accumulation rate (0.287 gL^−1^d^−1^) of fatty acids in the hydrolysate medium was considerably lower than that (0.882 gL^−1^d^−1^) in the glucose medium (Table 1), indicating a possible repression of the biosynthetic enzymes by inhibitory compounds in the *Enteromorpha* hydrolysate. Consequently, while the strain was able to produce an estimated maximum TFA concentration of 1.09 g/L on the glucose medium, it could produce 0.628 g/L on the hydrolysate medium. Nevertheless, our study revealed that *Enteromorpha* hydrolysate, which yielded up to 57.6% of the TFA content produced on glucose medium, could be a potential natural substrate for lipid production using thraustochytrids.

A previous study evaluated the potential of marine macroalgae as a sustainable source of renewable biomass for the production of single-cell oils (SCOs) through a biorefinery system [55]. The study specifically looked at the environmental and economic sustainability of producing SCOs from the seaweed *Saccharina latissima* using the oleaginous yeast *Metschnikowia pulcherrima*. The study found that seaweed-derived SCO lipids can be comparable to a terrestrial oil mix and suggests that seaweed offers a viable proposition for the competitive production of exotic oils. The results of our study support this previous research and provides evidence for the potential use of macroalgal hydrolyzates for the production of valuable fatty acids. 

### 2.3. Effect of Enteromorpha Hydrolysate Concentration on Growth and Fatty Acid Content 

The effects of various concentrations of *Enteromorpha* hydrolysate and glucose on the DCW and TFA content of the PKU#Mn4 strain were measured and compared in this study. The results revealed a significant effect of hydrolysate and glucose concentrations on both the DCW and TFA content (Figure 2). Among the tested concentrations of hydrolysate, 100 g/L provided the maximum DCW (4.32 ± 0.09 g/L) (Figure 2a) and TFA content (0.65 ± 0.03 g/L) (Figure 2b), while 20 g/L and 20–40 g/L of glucose provided the maximum DCW (5.98 ± 0.30 g/L) (Figure 2d) and TFA content (~1.0 g/L) (Figure 2e), respectively. In terms of TFA yield, 80 g/L of hydrolysate and 40 g/L of glucose provided the maximum values of 164 ± 16 mg/g DCW (Figure 2c) and 196 ± 10 mg/g DCW (Figure 2f), respectively. The TFA yield on *Enteromorpha* hydrolysate was comparable with those obtained on other reported feedstocks (Table 2). The lower TFA yield obtained in our study could be attributed to the use of the PKU#Mn4 strain, which showed a low TFA yield even in the glucose medium. Enhancing the lipid production of *Enteromorpha* hydrolysates by using more potential thraustochytrid strains could be an interesting topic of future investigation [56]. 

Macroalgae are a promising source of biologically active compounds with health benefits, such as polysaccharides and peptides [30]. Brown seaweeds are the most studied, but their polyphenol content can prevent fermentation [57]. In the present study, we demonstrated the feasibility of fermenting the biomass of green alga *Enteromorpha* to produce fatty acids. More importantly, fermentation of macroalgae waste can improve the release of bioactive compounds, but the target compound and algae characteristics determine the fermentation type and conditions [29]. Applying pretreatment can promote a higher release of bioactive compounds, but excessive treatment can lead to degradation and loss of bioactive properties [57]. More research is needed to understand the use of algae fermentation for food and nutraceutical applications.

Our results showed that the DCW tends to decrease when the hydrolysate concentration is raised above 100 g/L. Other studies using thraustochytrids also reported low DCW yield when the substrate concentration in the fermentation medium exceeded a certain level [23,58,59]. As *Enteromorpha* biomass is mainly constituted by sulfated polysaccharides, its acid hydrolysis might release sugars that may have inhibitory effect at higher concentrations. Further investigation on the structure and function of the released sugars and compounds upon acid hydrolysis is needed to understand the mechanism of inhibition at high substrate concentration. Furthermore, our experimental data also indicated a decline in the DCW and TFA content when the glucose concentration in the fermentation medium exceeded 40 g/L, which likely suggested glucose repression of growth and activity. Indeed, the fungus *Tuber borchii* growth is affected by high glucose concentrations, possibly because of an increased osmotic pressure [60].

Further compositional analysis of the TFA revealed equivalent fractions of saturated fatty acids (SFAs) and polyunsaturated fatty acids (PUFAs) in the TFA contents produced by PKU#Mn4 on hydrolysate or glucose medium (Appendix A). The SFAs were mainly C14:0, C15:0, C16:0, C17:0, and C18:0, while the PUFAs were C20:5, C22:5, and C22:6. When the PKU#Mn4 strain was cultivated with various concentrations of hydrolysate, C16:0 alone accounted for about 31.06 to 34.25% of TFA (Figure 3, Appendix A). However, when the strain was cultured in a glucose medium (Figure 3, Appendix A), C16:0 accounted for 31.88 to 41.12% of TFA. On the other hand, the fractions of C15:0, C17:0, and C18:0 derived from hydrolysate fermentation were much higher than those from the glucose fermentation. The occurrence of odd-chain fatty acids (mainly C15:0) may be due to the branched-chain amino acids present in the macroalgal hydrolysate as reported previously [7]. Furthermore, the total SFA content ranged between 42.18 g/L and 49.64 g/L in hydrolysate medium (Appendix A) and 37.15 g/L and 50.07 g/L in glucose medium (Appendix A). The maximum total SFA content was achieved at 120 g/L of hydrolysate medium, while the same was achieved at 30 g/L of glucose medium. 

**Table 2 marinedrugs-21-00198-t002:** Comparative biomass and TFA production by various thraustochytrid strains.

Strain	Carbon Source	Biomass (g/L)	TFA (g/L)	Maximum TFA Yield (g/g Biomass)	Reference
*Schizochytrium* sp. HX-308	Cane molasses	25.54	5.21	0.20	[22]
*Schizochytrium* sp. BCRC33482	Sugarcane bagasse	10.45	4.72	0.45	[15]
*Aurantiochytrium* sp. YLH70	Jerusalem artichoke	32.71	19.72	0.60	[16]
*Aurantiochytrium limacinum* SR21	Sweet sorghum juice (50%)	9.38	6.86	0.73	[17]
*Aurantiochytrium* sp. KRS101	Empty palm fruit bunches	34.40	12.50	0.36	[61]
*Aurantiochytrium limacinum* PKU#Mn4	*Enteromorpha* hydrolysate	4.32	0.65	0.16	This study

Among the PUFAs, the content of C22:6 (DHA) derived from hydrolysate fermentation (35.59–39.13%) (Appendix A) was slightly lower than that from glucose fermentation (37.60–43.26%) (Appendix A). Interestingly, the PKU#Mn4 strain yielded a much higher fraction (2.61–3.22%) of C20:5 (EPA) in the hydrolysate medium than that (0.25–0.49%) in the glucose medium. Previous studies have mostly focused on DHA; therefore, reports on EPA production are rare. Moreover, the EPA fraction only accounted for a small percentage of TFA (usually less than 1%) for most of the reported thraustochytrid strains [62]. Since EPA is also an important PUFA with high nutraceutical value, our study provides an interesting finding that *Enteromorpha* hydrolysate could be a potential substrate for EPA production using thraustochytrids.

EPA is an omega-3 fatty acid that is generally found in fish oil and some algae [56,63]. It has been shown to have several health benefits, including reducing inflammation, improving heart health, and potentially reducing the risk of certain types of cancer [64,65]. EPA may also be beneficial for mental health, as it has been shown to reduce symptoms of depression and anxiety [66]. Additionally, EPA may have benefits for eye health, cognitive function, and immune system function [67]. Interestingly, in the present study, we show for the first time that this high-value bioactive compound can be produced from macroalgal biomass via microbial fermentation using thraustochytrids. This suggests a new potential source of EPA production, which could have environmental benefits. However, more research is needed to understand the effectiveness and safety of EPA produced in this way for human consumption.

## 3. Materials and Methods

### 3.1. Enteromorpha Sampling

*Enteromorpha prolifera* samples were collected from the coastal waters of Qingdao, China, in August 2018, and washed three times with sterile distilled water before being transported to the lab for further treatment. The samples were spread evenly in a thin layer (0.1–0.5 cm) on a tray and air-dried in an oven for 5 days at 60 °C. The dried samples were stored in a glass desiccator for further processing. 

### 3.2. Preparation of Enteromorpha Hydrolysate Medium

The dried *Enteromorpha* samples were ground into powder and sieved through a 200-mesh sieve (75 μm × 200). *Enteromorpha* powder and sulfuric acid (98%, *w*/*v*; Jiangtian Chemical, Tianjin, China) were mixed in a ratio of 5:1 and diluted with sterile distilled water to achieve a final sulfuric acid concentration of ca. 1.0% (*w*/*v*) (Appendix A). The acid-pretreated solution was hydrolyzed at 121 °C for 60 min, and the resulting supernatant was adjusted to pH 7.0 with calcium hydroxide (Sigma-Aldrich, Munich, Germany) and a clear *Enteromorpha* hydrolysate was obtained by filtration. The hydrolysate was then concentrated on a Rotary Evaporator (SmarVapor RE-501, Dechem-Tech, Hamburg, Germany) and made to 1 L with sterile distilled water to achieve the desired hydrolysate concentration (40–120 g/L on a dry weight basis). 

### 3.3. Batch Fermentation

A previously isolated strain of thraustochytrid [68], renamed *Aurantiochytrium limacinum* PKU#Mn4 (CGMCC 8091), was used in this study. The strain was maintained on modified Vishniac’s (MV) agar plates at 28 °C and subcultured every 4 weeks [62]. The seed culture was prepared following the procedure described in our previous study [11]. Batch fermentation experiments were performed in 100 mL Erlenmeyer flasks with 50 mL of MV medium (1.5 g/L peptone (Oxoid, Hampshire, UK), 1 g/L yeast extract (Oxoid, Hampshire, UK), 0.25 g/L KH_2_PO_4_ (Kermel, Tianjin, China), and 33 g/L artificial sea salt (Yier, Guangzhou, China)). The pH of the MV medium was adjusted to 7.0. The MV medium was autoclaved at 115 °C for 21 min. Five mL of the seed culture was inoculated into the sterile MV medium. Batch experiments were set up in triplicates for each concentration of *Enteromorpha* hydrolysate (40–120 g/L on a dry weight basis) or glucose (10–50 g/L, Damao, Tianjin, China). *Enteromorpha* hydrolysate or glucose was added to the MV medium before autoclavation. All batch cultures were incubated at 28 °C under reciprocal shaking (150 rpm) for 4 days. Samples were collected from the batch culture at regular intervals for further analysis. 

### 3.4. Analytical Methods

The total reducing sugars was quantified using the dinitro salicylic acid (DNS, 9 Ding Chemistry, Tianjin, China) method [69]. For the DNS assay, 0.1 mL of the *Enteromorpha* hydrolysate was diluted with 1.9 mL double distilled water, and 0.5 mL of this diluted solution was added to 0.5 mL of DNS reagent, which was then incubated in a water bath at 100 °C for 5 min. After making the final volume of the incubated mixture to 10 mL, its absorbance was measured at 520 nm on a spectrophotometer (model #752, Jinghua, Shanghai, China). The content of the total reducing sugars was estimated based on a glucose standard curve. The yield of total sugar from *Enteromorpha* hydrolysate was calculated from the following equation.
$$Total\;reducing\;sugars\;(\%)=\frac{reduced\;sugar\;scontent\times volume\;of\;hydrolysate\times 0.9}{dry\;weight\;of\;Enteromorpha}\times 100$$

To identify the sugar composition of the *Enteromorpha* hydrolysate, the reducing sugars in the hydrolysate were derivatized with p-aminobenzoic acid (Sigma-Aldrich, Munich, Germany) and then separated on an HPLC (model #1260, Agilent, Santa Clara, Germany) equipped with a C18 076187 (250 × 4.6) column (Thermo Fisher Scientific, Shanghai, China). Briefly, a 100 μL of *Enteromorpha* hydrolysate was transferred into an Eppendorf tube containing 100 μL of p-aminobenzoic acid solution (0.7 g p-aminobenzoic acid (Sigma-Aldrich, Munich, Germany), 1 g acetic acid (Sigma-Aldrich, Munich, Germany), and 0.1 g sodium cyanoborohydride (Sigma-Aldrich, Munich, Germany) in 10 mL methanol (Macklin, Shanghai, China)). The tube was incubated in a water bath at 70 °C for 1 h. After incubation, the tube was cooled down to room temperature and the final volume of the reaction mixture was made to 1 mL with sterile distilled water. The sugar composition of the resulting solution was determined using HPLC. Isocratic elution with 5% methanol (Macklin, Shanghai, China) and 95% H_2_O for 30 min was employed. The UV wavelength, column temperature, and injection volume were 303 nm, 30 °C, and 20 μL, respectively. A similar procedure was followed for the standard solution of monosaccharides.

The dry cell weight (DCW) was quantified based on the gravimetric method described in our previous study [62]. The analysis of fatty acid composition was performed using the direct transesterification method [70] following the procedures detailed in our previous study [62]. 

### 3.5. Statistical Analyses

The mean, SD, and test of significance (ANOVA) for each parameter were computed in R software (version 4.0.0) [71]. A modified Gompertz model [72] was fitted to the experimental data to estimate the parameters: “*a*” (maximum growth potential, g/L), “*R*” (maximum growth rate, gL^−1^d^−1^), and “λ” (lag time, d) in R software (version 4.0.0). The data were plotted using Microsoft Excel and R package ggplot2 (https://ggplot2.tidyverse.org).

## 4. Conclusions

Use of natural substrates is an important research topic that is likely to continue to gain attention and interest in the future. In recent years, there have been several outbreaks of *Enteromorpha prolifera* in various coastal regions around the world, including in China, South Korea, and the United States. Our research indicates that mechanical equipment, such as dredges or harvesters, can effectively remove large quantities of algae in a short amount of time. This method is also cost-effective, as the cost of collecting *Enteromorpha prolifera* is negligible. Additionally, the removal of the algae biomass may receive incentives from the local government, as it contributes to the cleaning of coastal areas. This study reports the first successful fermentative production of PUFAs and SFAs from the hydrolysate of the marine macroalgae *Enteromorpha prolifera* collected from the coastal waters of Qingdao. The *Enteromorpha* hydrolysate contained a considerable amount of fermentable reducing sugars. Despite the lower growth rate and substrate consumption in the hydrolysate medium, the PKU#Mn4 strain was able to produce the maximum DCW and TFA content with an optimal hydrolysate concentration of 100 g/L. Interestingly, the C20:5n-3 fraction of the TFA content produced with hydrolysate medium was considerably higher than that with glucose medium. Taken together, this study provides a method to produce high-value fatty acids from a marine macroalgal hydrolysate using a thraustochytrid strain. The proposed method has the potential to benefit both the environment and industry.

## Figures and Tables

**Figure 1 marinedrugs-21-00198-f001:**
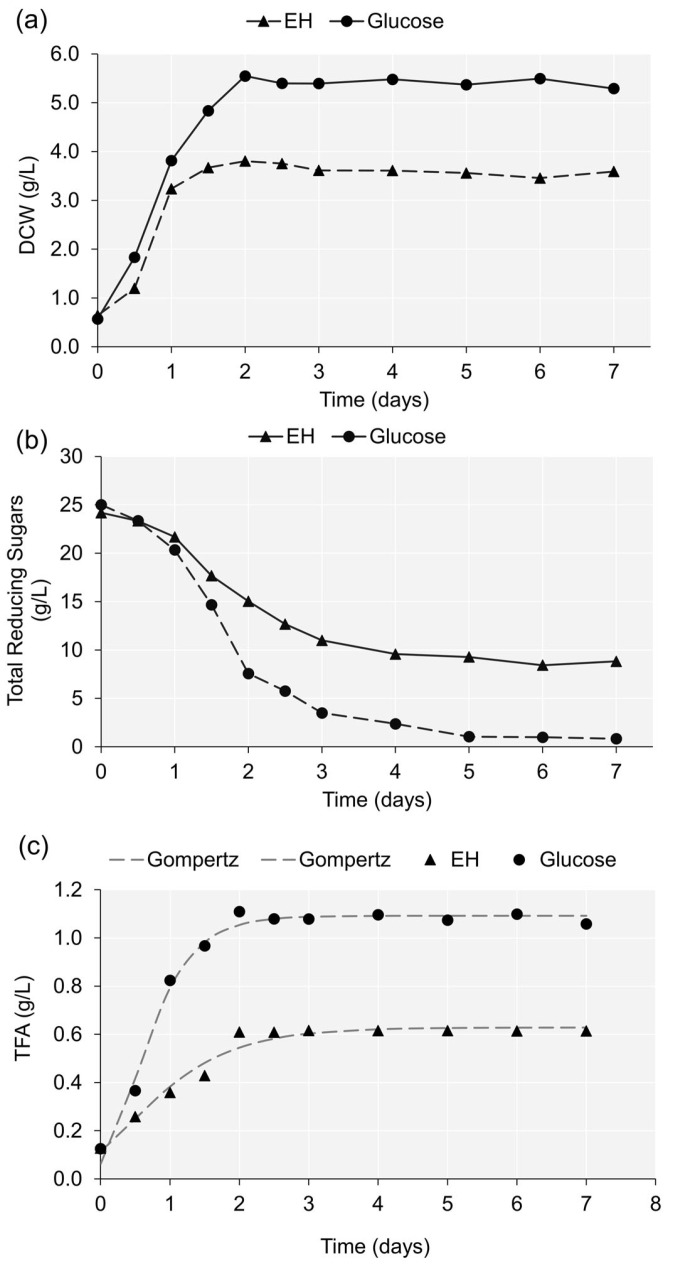
Time course of (**a**) DCW and (**b**) total reducing sugars, and (**c**) experimental and estimated (modified Gompertz model) total fatty acids (TFA) during the cultivation of the PKU#Mn4 strain on *Enteromorpha* hydrolysate and glucose media. EH stands for *Enteromorpha* hydrolysate.

**Figure 2 marinedrugs-21-00198-f002:**
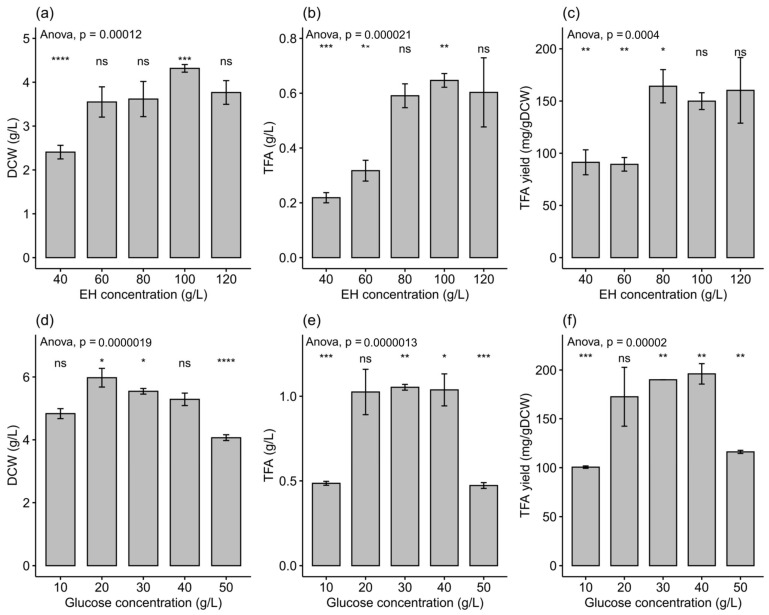
Effects of various concentrations of (**a**–**c**) *Enteromorpha* hydrolysate and (**d**–**f**) glucose on the DCW, TFA content, and TFA yield of PKU#Mn4 strain. The significant codes indicate the results of multiple comparisons, where the mean of each group was compared to all (i.e., base mean) by a paired *t*-test. The significant codes *, **, ***, and **** represented significance at *p* < 0.05, *p* < 0.01, *p* < 0.001, and *p* < 0.0001, respectively. The data represent the mean ± SD of triplicate samples (*n* = 3).

**Figure 3 marinedrugs-21-00198-f003:**
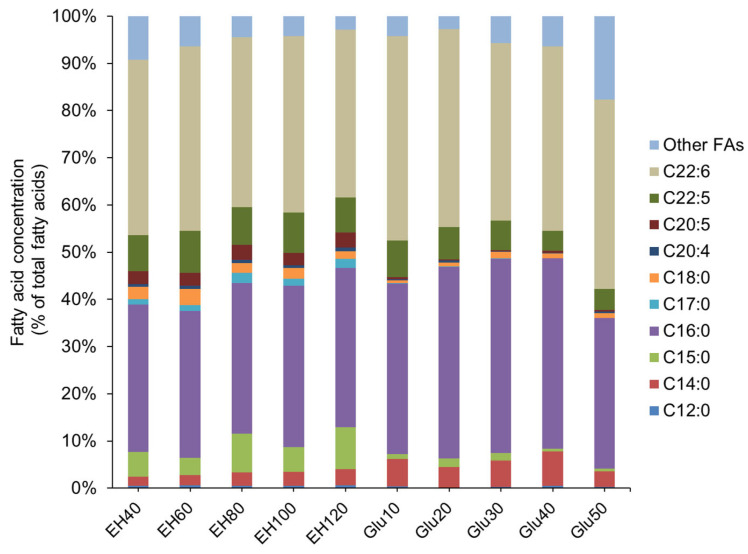
Effects of various concentrations of *Enteromorpha* hydrolysate (EH) and glucose (Glu) on the relative concentrations of fatty acids produced by PKU#Mn4 culture. The number after EH and Glu represents the concentration of EH and Glu.

**Table 1 marinedrugs-21-00198-t001:** Estimates of modified Gompertz model parameters after fitting experimental data.

Dependent Variable		*a* (g/L)	*R* (gL^−1^d^−1^)	λ (d)	Residual Standard Error
DCW	EH medium	3.635 ***	6.201 *	0.307 **	0.251
	Glucose medium	5.469 ***	4.183 ***	0.026	0.172
TFA	EH medium	0.628 ***	0.287 ***	−0.369	0.034
	Glucose medium	1.092 ***	0.882 ***	0.028	0.039

DCW: dry cell weight; TFA: total fatty acids; EH: *Enteromorpha* hydrolysate; *a*: maximum growth potential (g/L); *R*: maximum growth rate (gL^−1^d^−1^); λ: lag time (d); and significance codes: *** 0.001, ** 0.01, * 0.05.

## Data Availability

Not applicable.

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
