# Peer review of "Saturated and Polyunsaturated Fatty Acids Production by Aurantiochytrium limacinum PKU#Mn4 on Enteromorpha Hydrolysate"

_marinedrugs, 2023, doi:10.3390/md21040198_

Round 1

Reviewer 2 Report

The reviewer finds the paper interesting and relevant ( low cost substrate for fatty acid production), and the text is generally easy to follow.

However, there are few points that that can be improved:

Line 27: low cost – what is the cost? The reviewer would like to see some cost estimate.  Will this be at lower cost than crude glucose or cane-molasses or glycerol? This comes back in Results and discussions and Conclusion.

Line 159: predicted in not right, this value is found by calculation from the Gompetz function. “estimated”, “calculated or “found” could go.  Also line 184, 243 and other places should be changed.

Line 165: furfural is mentioned. what about poly-phenols, the hydrolysate was said to be clear but was it brown?

Line 170: explain in table caption DCW and TFA ( or under as for EH). Also what is a, R and alpha? (it is in M&M but should also be in the caption)

Line 229: include (EH) in caption for clarification.

Lines 261-265 (Figure 3)

             The corresponding figs (e.g. a and d) don’t have the same scale, this makes it difficult for the reader to compare

             What is n behind the error bars

             TFA is reported in g/L but mg/gDCW . Why not use g (or mg) for both?

Line 267: Table 3 is mentioned here but first appears in Material and methods (Line 309)

Line 286: It would be interesting if the authors could discuss the fact that from ~20g/L glucose consumed during the first 2-3 days “only” ~5g/L biomass is formed (see Figure 1). What happens to the rest, also as the N-part is important for biomass production.

Line 307-309: Table 3 and also in Supplements. In the table the long chain fatty acids having n-6, n-3….. However, in Table S2 this is not so. It could improve clarity if “n-?” would be added to Table S2

Line 314: Now room temperature can vary a lot. Was the temperature of the room controlled or reflecting outside temperature?

Line 317: was this yeast or yeast extract? Also, producers/supplier is missing. This applies for other chemicals and instruments.

Line 320: When was the EH added before or after autoclavation? This can be very important to explain as you have proteins and reducing sugars potential formation of Maillard reaction products that are inhibitory to microbial growth. If added after autoclavation how was the solution sterilized?

Line 324: The authors report analysis of sugars. What is the content of lipids and proteins/peptides in the EH?

Line 354: could the authors give the equation of the Gompertz function, and / or explain the modification?

Round 2

Reviewer 1 Report

The new version of the manuscript is highly improved and I would like to thank the Authors for the effort made to respond and apply most of my comments.

In the attached file please find a few comments I would like the authors to reply to.
